# Camptothecin Effectively Regulates Germline Differentiation through Bam–Cyclin A Axis in *Drosophila melanogaster*

**DOI:** 10.3390/ijms24021617

**Published:** 2023-01-13

**Authors:** Jing Zhang, Shijie Zhang, Zhipeng Sun, Yu Cai, Guohua Zhong, Xin Yi

**Affiliations:** 1Key Laboratory of Crop Integrated Pest Management in South China, Ministry of Agriculture, South China Agricultural University, Guangzhou 510642, China; 2Key Laboratory of Natural Pesticide and Chemical Biology, Ministry of Education, South China Agricultural University, Guangzhou 510642, China; 3Temasek Life Sciences Laboratory, National University of Singapore, Singapore 119077, Singapore; 4Department of Biological Sciences, National University of Singapore, Singapore 119077, Singapore

**Keywords:** camptothecin, germ cells, Bam, cell cycle

## Abstract

Camptothecin (CPT), first isolated from Chinese tree *Camptotheca acuminate*, produces rapid and prolonged inhibition of DNA synthesis and induction of DNA damage by targeting topoisomerase I (top1), which is highly activated in cancer cells. CPT thus exhibits remarkable anticancer activities in various cancer types, and is a promising therapeutic agent for the treatment of cancers. However, it remains to be uncovered underlying its cytotoxicity toward germ cells. In this study we found that CPT, a cell cycle-specific anticancer agent, reduced fecundity and exhibited significant cytotoxicity toward GSCs and two-cell cysts. We showed that CPT induced GSC loss and retarded two-cell cysts differentiation in a niche- or apoptosis-independent manner. Instead, CPT induced ectopic expression of a differentiation factor, bag of marbles (Bam), and regulated the expression of cyclin A, which contributed to GSC loss. In addition, CPT compromised two-cell cysts differentiation by decreasing the expression of Bam and inducing cell arrest at G1/S phase via cyclin A, eventually resulting in two-cell accumulation. Collectively, this study demonstrates, for the first time in vivo, that the Bam–cyclin A axis is involved in CPT-mediated germline stem cell loss and two-cell cysts differentiation defects via inducing cell cycle arrest, which could provide information underlying toxicological effects of CPT in the productive system, and feature its potential to develop as a pharmacology-based germline stem cell regulation agent.

## 1. Introduction

Camptothecin (CPT) is a pentacyclic alkaloid that was first isolated from stem wood of *Camptotheca acuminate* [1]. The US National Cancer Institute screening program identified CPT as a promising therapeutic agent for the treatment of cancers because it specifically targets topoisomerase I (top1), which is highly activated in cancer cells [2]. CPT could prevent the re-ligation of the nicked DNA and dissociation of top1 from the DNA by binding to both of the top1 enzyme and the intact DNA strand through hydrogen bonding. During replication, this CPT-involved ternary complex could act as a roadblock for the replication fork to result in shear stress upon the intact DNA strand, and eventually leading to breakage, DNA damage, and cell death [3,4,5]. Previous pharmacological studies indicated that CPT could inhibit DNA and RNA (including ribosomal RNA) synthesis, induce DNA damage, and arrest cell cycle at both S and G2 phases [6,7,8]. To this end, CPT could induce G1/S phase arrest in oral squamous cancer cells, ovarian clear cell carcinoma, and was found to possess a wide spectrum of antitumor activities [9,10,11]. However, the severe side effects of CPT, such as nausea, vomiting, dermatitis, diarrhea, cystitis, leukopenia, precluded its initial clinical development as a chemotherapeutic drug [12]. Despite many pharmacokinetic studies, it remains to be unpredictable underlying its toxicity toward different systems.

*Drosophila* ovarian system has been considered as a fruitful in vivo system for studying cell biology and dissecting the cytotoxicity of CPT in productive system [13]. In females, the gametes are produced from a specialized tissue called germline stem cells (GSCs) [14]. As the source of gametes, the only cell type that can pass the genetic information to the next generation, GSCs play a fundamental role in maximizing the quantity of gametes that animals produce, while ensuring their highest quality [14]. The maintenance and differentiation of GSC is modulated by intrinsic and extrinsic signal pathways in the ovary to be instructive to specify cell fate [15]. As one of those identified intrinsic factors to be able to regulate GSCs [16,17,18,19], cell cycle control serves as a critical aspect in the decision between GSC maintenance and differentiation [20]. The new GSC daughter and the cystoblast (CB) remain connected throughout most of the cell cycle; consequently, changes to the cell cycle would be critical for promoting or hampering GSC differentiation [21]. For instance, loss of cyclin B (CycB), a late-G2 phase regulator to control the G2-M phase transition, could lead to GSC self-renewal and maintenance defects [20]. Additionally, following gamma irradiation, GSCs briefly pause the cell cycle and become ‘quiescent’, resulting in progeny loss [22,23]. Although a plethora of intrinsic factors have been identified for their roles in regulating stem cell fate via affecting the cell cycle [22,24], it remains largely unclear whether CPT, a cell cycle-specific anticancer agent, could affect GSC fate.

In this study, we unveiled the cytotoxicity of CPT in GSC loss and two-cell differentiation defects. Instead of niche and apoptosis, the differentiation factor, bag of marbles (Bam), directly received CPT signals to influence GSC numbers. In addition, CPT compromised germ cell differentiation by decreasing the expression of Bam and inducing cell arrest in G1/S phase via cyclin A (CycA), resulting in two-cell accumulation. Overall, this study demonstrates, for the first time in vivo, that the Bam–CycA axis is involved in CPT-mediated GSC loss and two-cell differentiation defects via inducing cell cycle arrest, and the results in this study highlight the toxicological role of CPT in germ cells and its potential to serve as a pharmacology-based GSC regulation agent.

## 2. Results

### 2.1. The Effects of CPT on Germ Cells

CPT showed strong cytotoxicity against a variety of tumor types in vitro and in vivo; we then sought to examine the mortality of flies following CPT treatment. We found that although CPT-treatment did not have noticeable impact on fly survival rate, the treated females produced significantly fewer eggs, compared to control-treated females (Figure 1a,b), suggesting that CPT may affect fly oogenesis. Indeed, while the ovariole of the control female contained 5–7 follicles, the CPT-treated female only possessed 1–2 follicles at the discontinued stage (Figure 1h). In addition, some phenotypes were also observed at the anterior tip of the ovariole—the germarial region in treated flies, which is the focus of this study. In the germarium, two to three GSCs (identified by pMad-positive cells) resided at the anterior tip, next to a cluster of cap cells [25], and contained an anteriorly-positioned spherical fusome (or spectrome) (Figure 1c), but CPT treatment led to a reduction in the number of GSCs (Figure 1d). The daughters of GSCs, namely cystoblasts (CBs), move posteriorly as they differentiate: each CB divides four times with incomplete cytokinesis to form an interconnected 16-cell cyst in which one of the cells adopts oocyte fate and the rest become supporting nurse cells [26]. We found that CPT also affected the early steps of the germ cell development process and blocked CB differentiation, resulting in accumulation of CB-like (Figure 1e,f) and two-cell (Figure 1g,h) in the germarium. Each cyst could be surrounded by a single layer of follicle cells to form egg chambers. However, CPT-treated flies possessed a significantly higher ratio of abnormal phenotypes of egg chamber (apparent failure of egg chamber budding leading to empty ovarioles) compared with control flies (Figure 1i,j). Those observations indicated that CPT treatment led to a GSC loss and also a delay in germ cell differentiation, consequently resulting in defects in egg production.

### 2.2. CPT Treatment Fails to Cause Apoptosis and Influence Niche in Ovarian GSCs

In the *Drosophila* ovary, extrinsic signaling from niche and intrinsic translational machinery regulate the balance between GSC maintenance and differentiation of their daughters [27]. We investigated whether CPT-induced GSC loss was due to apoptosis by examining the expression of two cell death markers, including Terminal deoxynucleotidyl transferase-mediated dUTP Nick End Labeling (TUNEL) assays and cleaved Caspase-3 activities. We found no cleaved TUNEL signals and Caspase-3 activity detected in GSCs, CBs-like and two-cell, in both control and CPT treated ovaries (Figure 2a,b), indicating that the defects in GSC maintenance by CPT treatment are not a result of cell death. We next addressed whether the GSC loss is a consequence of premature differentiation. At the anterior tip of the ovary, terminal filament (TF) cells, cap cells (Cpc), and escort cells (ECs) could provide a physical location to house GSCs and form a GSC niche, which could send signals to GSCs, including Decapentaplegic (Dpp), Hedgehog (Hh), and Unpaired (Upd), to regulate their proliferation to maintain tissue homeostasis [28,29]. As the spatial organization of the GSC niche permits direct contacts between two or three CpCs and one GSC, which are anchored to the CpCs by adherent junctions, we then first examined whether the GSC loss induced by CPT attributed to the defects in Dpp expression by using *dpp-lacZ* in combination with anti-Engrailed (En) antibody to mark En and Dpp expression in Cpcs. We found that neither the protein accumulation of En nor the level of Dpp (Figure 2c) showed significant changes after CPT treatment. Since the expression of Dpp was unaffected, the number of Cpcs was measured by using anti-lacZ staining, and the results showed the number of Cpcs remained still after treatment (Figure 2e and Appendix A), indicating CPT did not induce changes in Cpc number as well. We further confirmed the expression of *dpp* by in situ hybridization, which also showed no significant changes were observed at transcript level following CPT treatment (Figure 2d). In addition, armadillo (Arm) is concentrated at the interface between Cpcs and GSCs in the adult ovary, which supports a role of this adhesion system in anchoring GSCs to their niche [30,31]. Still, we found no changes in Arm expression level and pattern after CPT treatment (Figure 2f), suggesting CPT induced decline in the number of GSCs was independent of Arm. Together, these results suggest that CPT-induced GSC loss is likely not a result of defective niche activity. We then investigated how CPT-treatment affected CB and two-cell cyst differentiation. As EC-expressed Thickveins (Tkv) acts as a receptor sink to remove excess Cpc-expressed Dpp, thereby promoting GSC differentiation [32], we then examined Tkv expression by immunostaining with anti-Tkv to address whether the levels of Tkv in the germarium were affected by CPT and accounted for GSC loss. The results showed that similar fluorescence intensity and identical distribution patterns were observed in fly ovaries with or without CPT treatment, indicating CPT treatment could not trigger expression changes of Tkv at both protein and transcript level (Figure 2g). As EC cellular processes are also closely associated with differentiated germ cells, and the physical interactions between ECs and germ cells are essential for GSC differentiation [26], and Erk signaling has an important role in generating EC shapes and protrusive activity, we then examined the pErk activity in EC after exposure to CPT. The results showed that pErk was readily expressed in control EC, and CPT treatment did not induce significant changes in pErk expression. We noted that the expression of pErk increased in the early follicular cell, which unlikely contributed to GSC loss (Figure 2h). Moreover, we further investigated whether ECs could be influenced by CPT by using *pz1444* and anti-3A9 to identify ECs in the anterior region where GSCs are localized. For both control and CPT treated groups, we could observe positive signals for ECs in the region 1 or 2a (*n* = 83), while after CPT treatment, the localization of ECs expanded to the posterior regions (i.e., egg chamber) (Figure 2i). However, such abnormal phenotype of EC localization unlikely contributed to GSC loss. We thus excluded the roles of apoptosis and niche in CPT-induced GSC loss and differentiation defects.

### 2.3. Bam Signal Was Involved in CPT-Induced Toxicity in Germarium

As Bam serves as a key factor to regulate GSC differentiation and germline cyst development, we then further examined whether Bam signaling was involved in CPT-induced GSC loss and two-cell accumulation. Germaria staining with Bam indicated that CPT treatment induced ectopic expression of Bam in GSCs, but suppressed the high expression of Bam in differentiation cells. In some germaria, the protein expression of Bam expanded to the posterior end of the tumorous germarium (Figure 3a). These results were further confirmed by staining with a Bam-GFP (GFP gene driven by the Bam promoter) reporter, which also showed ectopic GFP signals in GSCs but weaker fluorescence intensity in differentiating germ cells in CPT-treated germarium compared with control (Figure 3b,c). The transcriptional expression of *bam* was further confirmed by in situ hybridization. Consistent with changes in protein level, CPT induced ectopic *bam* expression in GSCs but decreased mRNA level of *bam* in differentiating germ cells (Figure 3d). To verify the role of Bam in CPT-induced two-cell accumulation, we checked the effects of Bam mutation on the phenotype of germaria with or without CPT treatment. Germarium bearing one copy of *bam^86^* mutation contained a slight increase in the number of two-cell compared with wild type flies. Interestingly, CPT-treatment led to a drastic increase in CB and two-cell cyst in the germarium with copy *bam^86^* mutation, leading to typical tumor-like germarium (Figure 4a). We thus recorded the number of two-cells, and found that CPT exposure in *bam^86^* mutation germarium triggered an increase in the number of two-cells by 3.2-fold, compared with *bam^86^* mutation germarium without CPT treatment (Figure 4b). We speculated that CPT could induce DNA damage and cell cycle arrest in two-cell phase by suppressing the expression of Bam to delay the differentiation of CB and two-cell cysts. To further determine whether increasing Bam expression could promote two-cell differentiation, we used the *hs-bam* transgene, in which Bam expression is under control of the heat-shock-inducible *hsp70* promoter to drive Bam expression in the ovaries, with or without CPT treatment. CPT-treated *hs-Bam* transgene germaria, without heat-shock, contained more two-cell cysts compared with DMSO-treated *hs-bam* transgene germaria without heat-shock (Figure 4c), indicating that the heat shock *bam* construct itself does not affect two-cell differentiation. Heat-shock-induced Bam expression with CPT treatment can sufficiently promote the differentiation of two-cell cysts into four-cell or eight-cell cysts, when compared with control flies of *hs-Bam* transgene germaria without heat shock (Figure 4d). These results demonstrated that CPT could influence the differentiation process by regulating Bam expression.

### 2.4. The Role of Top1 in CPT-Induced Toxicology 

To examine if top1 was also required for CPT-induced toxicology in GSCs, we used *nos-gal4*-driven RNAi expression to knock down *top1* gene expression specifically in germ cells. Our results showed that germline-specific *top1*^RNAi^ shrunk the germarium and led to rapid germ cell loss, and that CPT treatment could exacerbate such GSC loss phenotype, exhibiting similar but more severe abnormal germarium and triggering almost entire GSC and CB loss (Figure 5a,b). Control germaria maintained two or three GSCs and two CBs, and *top1*^RNAi^ germarium contained 1.4 GSCs and 1.5 CBs, respectively; however, *top1*^RNAi^ germarium with CPT treatment did not carry any GSCs or CBs (Figure 5c,d). The severity of GSC and CB loss phenotype in *top1*^RNAi^ germarium with CPT treatment suggested that top1 and CPT work in a linear pathway. Unexpectedly, germline-specific *top1* knockdown with CPT treatment also led to significant two-cell cyst loss compared with *top1*^RNAi^ germarium without CPT treatment (Figure 5e), which might be attributed to germ cell loss in some cases from *top1*^RNAi^ germarium with CPT treatment. We recounted the number of two-cell cysts by excluding the cases with complete germ cell loss, and found that *top1*^RNAi^ germarium with CPT treatment could lead to significant accumulation of two-cell, compared with *top1*^RNAi^ germarium without CPT treatment (Figure 5f). To further illuminate the relationship between top1 and the aforementioned differentiation factor, Bam, in CPT-induced GSC loss and two-cell accumulation, we examined Bam expression in *top1*^RNAi^ GSCs with or without CPT treatment. The results showed that CB and two-cell cysts expressed readily detectable levels of Bam; however, ectopic Bam expression occurred in a number of GSCs in *top1*^RNAi^ germarium without CPT treatment, or phenocopied CPT treatment. These data suggested that changes in top1 expression were sufficient to induce ectopic Bam expression in GSCs. CPT exposure in a *top1*^RNAi^ background also induced ectopic expression of Bam in GSCs (Figure 5g). These results indicated that top1 was involved in CPT-induced ectopic Bam expression, GSC loss, and two-cell accumulation. As DNA damage might be induced by CPT treatment via targeting top1, we examined p53 activity in early germ cells as its activity in *Drosophila* ovarian GSCs could be activated in response to DNA damage [33,34]. We could not find any obvious p53 signal in control of GSC, CB, and two-cell cysts, while CPT treatment could induce significant upregulation of p53 activities as evident by upregulated GFP signals in GSC, as well as in two-cells in CPT-treated germarium (Figure 5h). These data suggested that different degrees of DNA damage occurred in early germ cells in response to CPT treatment.

### 2.5. CPT Treatment Led to Cell Cycle Arrest

To monitor cell cycle progression of germ cells in the germarium, we used immunofluorescence combined FUCCI (fluorescent ubiquitylation cell cycle indicators) to identify cells in different phases of the cell cycle [35]. We then expressed *UASp-GFP-E2f1* (1–230) *UASp-mRFP1-cycB* (1–266) under control of *nos-gal4* to distinguish G1, S and G2 phases of interphase (Figure 6a). For CB in the control group, the fraction of proliferating G2/M and S cells accounted for 94.05% and 3.57% (*n* = 84), and after CPT treatment, the proportion of G2/M and S cells changed to 55.96% and 25.69%, respectively (*n* = 109) (Figure 6b). The same situation also occurred in two-cell, as the proportion of G2/M and S cells fluctuated from 87.95% and 10.84% (*n* = 83) to 48.26% and 28.36% (*n* = 201), respectively (Figure 6b). Additionally, the number of GSCs in phase G2/M transition accounted for 99.26% (*n* = 135) in the control group, and exhibited no significant change after exposure to CPT (the proportion of cells in phase G2/M was 90.48% in CPT treated group, *n* = 63) (Figure 6b). Consistent with a previous study, during cell division, high intensities of green and red fluorescence could be observed when nuclear envelope breakdown and degradation of the red probe (mRFP-NLS-CycB_1–166_) occurred. Based on this fact, the intensity of the green signal (GFP-E2F1_1–230_) would drop dramatically (as CRL4^Cdt2^ is activated at the G1-S transition) after a ~10 h steadily increase. After ~1 h gap period (without signal), the red fluorescence intensity would increase followed by reaccumulation of the green probe [35]. Our results showed that following CPT treatment, both the CycB and E2f1 signals were absent in 17.43% and 22.89% cells from CB-like and two-cell, significantly higher compared with control (2.38% and 1.20% from CB and two-cell in control group) (Figure 6b), which indicated that CPT treatment led to cells undergoing phase G1/S or S arrest, and such arrest might contribute to the previously observed increase in the number of two-cells after CPT treatment.

### 2.6. CycA Was Involved in CPT-Induced Differentiation Defects

To illuminate the roles of cell-cycle regulators in CPT-induced differentiation defects, we examine the effects of CycA, cyclin B (CycB), and cyclin E (CycE) overexpression and RNAi on germ cells following CPT exposure. The results showed that overexpression of CycA with CPT exposure could enhance the toxicity of CPT exposure, exhibiting a higher rate of GSC loss and more two-cell accumulation (Figure 7a). These findings were consistent with a previous study, where ectopic expression of Bam in GSCs could be enhanced by co-expression of CycA [36]. However overexpression of *cycB* (Figure 7b) or *cycE* RNAi (Figure 7c) did not display expanded undifferentiated cell phenotypes. As one of the downstream targets of Bam in GSC [24], these results suggested that CycA expression might be involved in CPT-induced GSC loss and two-cell accumulation.

To further determine whether CycA is the direct cause of GSC loss and two-cell accumulation, we used *nos-gal4* to drive *CycA* gene overexpression specifically in germ cells. Immunostaining showed that *CycA* overexpression displayed a more severe phenotype of GSC loss in the germarium compared with CPT treatment (Figure 8a,e,f). However, *CycA* overexpression did not show a significant increase in the number of two-cell cysts (Figure 8b,e,f), which might be attributed to low efficiency of *nos-gal4*-driven overexpression in two-cell cysts and phenotype of germ cell loss induced by overexpression of *CycA*, overriding the two-cell accumulation phenotype. To confirm these observations, we used *nos-gal4* to drive down-regulation of the *CycA* gene specifically in germ cells. The genetic experiments revealed that the decreasing of *CycA* with CPT treatment can significantly restore GSC number, indicating that the down-regulation of *CycA* expression can rescue GSC loss phenotype caused by CPT treatment (Figure 8c,g,h). Moreover, the decline in the level of *CycA* can also abrogate two-cell accumulation induced by CPT (the number of two-cell cysts in the control group was 3.52, and the number in *CycA* RNAi germarium was 2.91) (Figure 8d,g,h). These results further confirmed that CycA participated in the CPT-induced GSC loss and two-cell accumulation.

## 3. Discussion

It is well established that CPT-induced toxicity is dependent upon top1, which is capable of introducing a transient single-strand break in DNA, through which another strand can pass, thereby reducing DNA supercoiling [37,38]. Due to the specific function of top1, the single-strand cleavage/rejoining activity of top1 suggests that it may serve as a swivel for unwinding and rewinding of DNA helices associated with many critical cellular processes, including DNA replication, transcription, repair, and recombination induction of cell cycle arrest [39,40]. Therefore, CPT and CPT-derived chemicals were reported to cause severe toxicity in the hematopoietic system, lymphatic tissue, gastrointestinal tract, and reproductive organs, and have been developed for cancer therapy [41,42,43]. Our data demonstrated that CPT is deleterious to *D. melanogaster* germline development, and it leads to GSC loss and concomitantly blocks CB and two-cell cyst differentiation by ectopically inducing Bam expression and regulating CycA. Other than growing evidence on intrinsic factors-dependent mechanisms regulating the GSC lineage in *D. melanogaster* [44], we reported the toxicological effects of CPT on GSCs and two-cells for the first time, highlighting the potential of its pharmacophores to be developed in stem cell-based therapies.

It is well documented that top1 is the only cellular target of CPT [2]. Top1 activity is robust in malignant cells and correlates with disease progression in colorectal and ovarian cancers [45], making CPT a potent agent for anticancer chemotherapy. We found that the inhibition of the expression of *top1* led to a large number of germ cell loss without CPT treatment, and the phenotype is similar to that of CPT treatment. At the same time, *top1^RNAi^* flies with CPT treatment could exacerbate GSC loss and two-cell accumulation phenotypes induced by CPT treatment alone, exhibiting more severe abnormal germarium and triggering almost the entire GSC and CB loss. Therefore, we suggested CPT affected GSC maintenance and two-cell accumulation through inhibiting the expression of *top1*. Based on the current study, it is noteworthy that we found oral ingestion of CPT led to GSC loss in *D. melanogaster* probably via interfering with the cell cycle, which serves as a critical aspect in the decision between GSC maintenance and differentiation. Our results suggest that during germline differentiation, CPT-mediated changes in CycA expression have a role in reprogramming self-renewal, leading to precocious GSC differentiation, and eventually contributing to GSC loss. A previous study indicated that overexpression of a stable form of CycA led to severe *Drosophila* GSCs loss [46], and stabilized CycA could prevent exiting from the cell cycle and entering into G1 at an appropriate developmental stage [47]. Thus, CycA plays an important role in cell cycle in *Drosophila* GSCs, and we also found CycB overexpression has no detectable phenotype in GSCs. In addition, we found CycA regulated GSC differentiation in a Bam-dependent manner, which is evident by the results that ectopic expression of Bam in GSCs would increase the stability of CycA, and down-regulation of CycA antagonized the function of ectopic Bam in GSCs. These results suggested that changes in CycA level are sufficient to explain the loss of GSCs when expressing Bam ectopically. Consistent with the previous study, they found that ectopic expression of the stable form of CycA in germ cells caused GSC loss, which is similar to the phenotype resulting from ectopic expression of Bam in GSCs [36,48]. Ji et al. convincingly showed that ectopic expression of Bam in GSC could be enhanced by co-expression of CycA, and suppressed by CycA reduction [24]. Furthermore, they found CycA can be coimmunoprecipitated with Bam from S2 cells and ovarian extracts [24,46,49]. The relationship between Bam and CycA might be interpreted as the way that Bam functions as a ubiquitin-associated protein to deubiquitinate and stabilize CycA, thereby balancing GSC self-renewal [24]. The results in this study indicated that the Bam–CycA regulatory axis plays an important role in GSC differentiation and cell-cycle alterations in response to the cytotoxicity of CPT, but the detailed regulatory mechanism remains to be further elucidated.

In the female germline, Bam is a key intrinsic regulator of differentiation [50] because Bam RNA appeared shortly after the differentiation of a stem cell that produces new CBs [51]. In the absence of Bam activity, GSC daughters failed to differentiate, and ectopic expression of Bam in GSCs was sufficient to induce GSC loss and led to the accumulation of undifferentiated germ cell tumors [52,53]. In our study, CPT treatment also resulted in accumulation of two-cell cysts, and such differentiation defects were further enhanced by the heterozygous mutation of *bam*. By contrast, heat-shock-induced Bam expression can sufficiently promote the differentiation of two-cell cysts, while reducing CycA can rescue two-cell accumulation caused by CPT treatment, indicating the Bam–CycA axis might also contribute to the two-cell accumulation phenotype. Previous study revealed the *bam* gene was required for the differentiation of CBs from the stem cells, perhaps by altering the cell cycle and stabilizing differentiation factors, such as CycA [51]. Moreover, Bam-dependent deubiquitinase complex can disrupt GSC maintenance by targeting CycA, and DNA damage could disrupt Bam-dependent differentiation pathways and cause the accumulation of CB-like cells in a Lok-dependent manner [54]. Based on those facts, we hypothesized that CPT treatment could regulate the expression of Bam, and the alterations in Bam expression are sufficient to trigger the transcription program of the cysts following differentiation, leading to cell arrest at G1/S and two-cell accumulation probably via affecting CycA.

## 4. Materials and Methods

### 4.1. Drosophila Stocks

*D. melanogaster* was reared by standard *Drosophila* medium. All fly stocks were maintained at 25 °C and a related humidity of 60% with 1:1 (light:dark) photoperiod. For each treatment group, newly emerged adults (<24 h) were placed into a vial containing standard media and were applied 100 mg/L CPT for 9 days. DMSO treated flies were considered as the control. The genotypes of the mutant lines used in this study were: *w1118* (used as wild-type control), *dpp2.0-lacZ* [55], *Bam-GFP* [50], *pz1444*, *nos-gal4 vp16*, *hs-bam* (a gift from Yu Cai [Temasek Life Sciences Laboratory, National University of Singapore, Singapore), *uas-top1*^RNAi^ (Bloomington, BL#55314), *bam^86^* (Bloomington, BL#5427), FUCCI (*UASp-GFP-E2F11–230*; *UASp-mRFP1-cycB1–266*, Bloomington, BL#55101), *hs-cycA* (Bloomington, BL#91660), *hs-cycB* (Bloomington, BL#91664), *UAS-cycE*^RNAi^ (Bloomington, BL#29314). All crosses were maintained at room temperature. For heat-shock stock, crosses were maintained at 25 °C. The flies were heat-shocked for 1 h at 37 °C following CPT treatment for 9 days, followed by 24 h recovery at room temperature. Additionally, the dissection and immunostaining were performed.

### 4.2. Survival Analysis 

Twenty adult flies (male:female = 1:1, 24 h post emergence) were placed in vials with or without CPT and maintained at 25 °C. The number of flies was counted every day and the mortality was calculated after all the flies died.

### 4.3. Fecundity Examination

Twenty adult flies (male:female = 1:1, 24 h post emergence) were placed in vials with or without CPT and the number of eggs laid at the 9th d post treatment was collected and counted. The egg production was measured within 24 h and three independent trials were performed.

### 4.4. Immunostaining

After the ovaries of female flies were dissected in PBS, the tissues were fixed with 4% paraformaldehyde (PFA) for 20 min at room temperature, rinsed with PBT (0.1% Triton X-100 in PBS) three times for 10 min each, blocked in 5% NGS (Normal goat serum) for 1 h, and incubated with primary antibodies overnight. Then, the samples were washed three times with PBT for 10 min each. After incubation with secondary antibody for 3 h, the tissues were stained with Hoechst for 10 min, and washed again with PBT 3 times.

The primary antibodies used in this study were listed as follows: rabbit anti-pMad (1:800; Cell Signaling Technology, Danvers, MA, USA), rabbit anti-Caspase 3 (1:2000, Cell Signaling Technology, USA), rabbit anti-pErk (1:400; Cell Signaling Technology, USA) mouse anti-lacZ (1:10,000; abcam, Cambridge, UK), and rabbit anti-LacZ (1:10,000; abcam, UK). Other antibodies were purchased from Temasek Life Sciences Laboratory, including mouse anti-3A9, rabbit anti-α-Spectrin [56], mouse anti-Bam, chicken anti-GFP, rabbit anti-Tkv [32], guinea pig anti-Vasa, mouse anti-Arm, mouse anti-En, rat anti-Fluorescein (FITC), Cy3- and Cy5-goat against rabbit, mouse, chicken, rat and guinea pig secondary antibodies were purchased from Jackson Immuno Research Laboratories (West Grove, PA, USA), Inc. The DNA dye used was Hoechst 33258 (1:5000; Cell Signaling Technology, USA). Samples were analyzed with an upright confocal microscopy.

For TUNEL (Roche, # 12156792910, Penzberg, Germany), the ovaries were dissected in PBS, and fixed in 4% PFA for 1 h. After washing three times with PBS for 10 min each, the tissues were incubated in Permeabilization solution for 2 min on ice. In total, 5 μL of enzyme solution was added to the 45 μL label solution to obtain 50 μL TUNEL reaction mixture, and this was shaken for 1 h at 37 °C, and then we continued with immunostaining procedures as mentioned previously.

### 4.5. In Situ Hybridization

For in situ hybridization, the probes were labeled by Roche DIG RNA labeling kit (Roche, #11175025910, Germany) following instructions of the manufacturer. Ovaries were dissected in PBS, and then fixed in 4% PFA overnight. After washing 3 times with PBT (PBS + 0.1% Tween 20), the tissues were again washed with methanol/PBT for 5 min and rinsed three times with PBT. After the samples were rinsed with 1:1 hybridization buffer/PBT for 5 min, 100% hybridization buffer for 5 min, and three times with PBT for 5 min each, respectively, the DIG-labeled RNA probes were pre-hybridized at 100 °C for 1 h prior to hybridization. For the hybridization, the tissues were incubated overnight with a probe at 60 °C. After hybridization, to wash off the unspecific binding, the tissues were rinsed with washing buffer four times for 30 min, and then washed with MABT buffer two times for 10 min. After blocking with 5% blocking solution, the tissues were incubated with anti-DIG-POD (1:200; Roche) in PBT (with 0.5% blocking solution) overnight. After washing with MABT for 1 h, we added 1 μL diluted fluo-dye (Roche) in amplification buffer into the tissue solution and kept it at room temperature for 1.5 h. Following the procedures of in situ hybridization, the immunostaining was carried out as previously mentioned [28]. Observations were carried out with an upright confocal microscopy.

### 4.6. Statistical Analysis

The data analyses were performed using SPSS software. The differences between two samples were analyzed by Student’s *t*-test (* *p* < 0.05, ** *p* < 0.01, *** *p* < 0.001, **** *p* < 0.0001).

## 5. Conclusions

In this study, we unveiled the cytotoxicity of CPT in GSC loss and two-cell differentiation defects, which could provide information for its therapeutic application. CPT could induce ectopic expression of Bam in GSCs via top1, and such a phenotype could be enhanced by overexpression of CycA, which might contribute to the observed GSC loss. In addition, CPT can cause DNA damage in the early germline cell by regulating Bam expression at both transcript and protein level, thus leading to cell arrest at G1/S and two-cell accumulation. Collectively, the results in this study provided convincing results that CPT may have therapeutic potential as an anticancer agent in germ cells. Further study is needed to evaluate the safety of CPT in advanced models to confirm the mechanism in germline cells of other organisms.

## Figures and Tables

**Figure 1 ijms-24-01617-f001:**
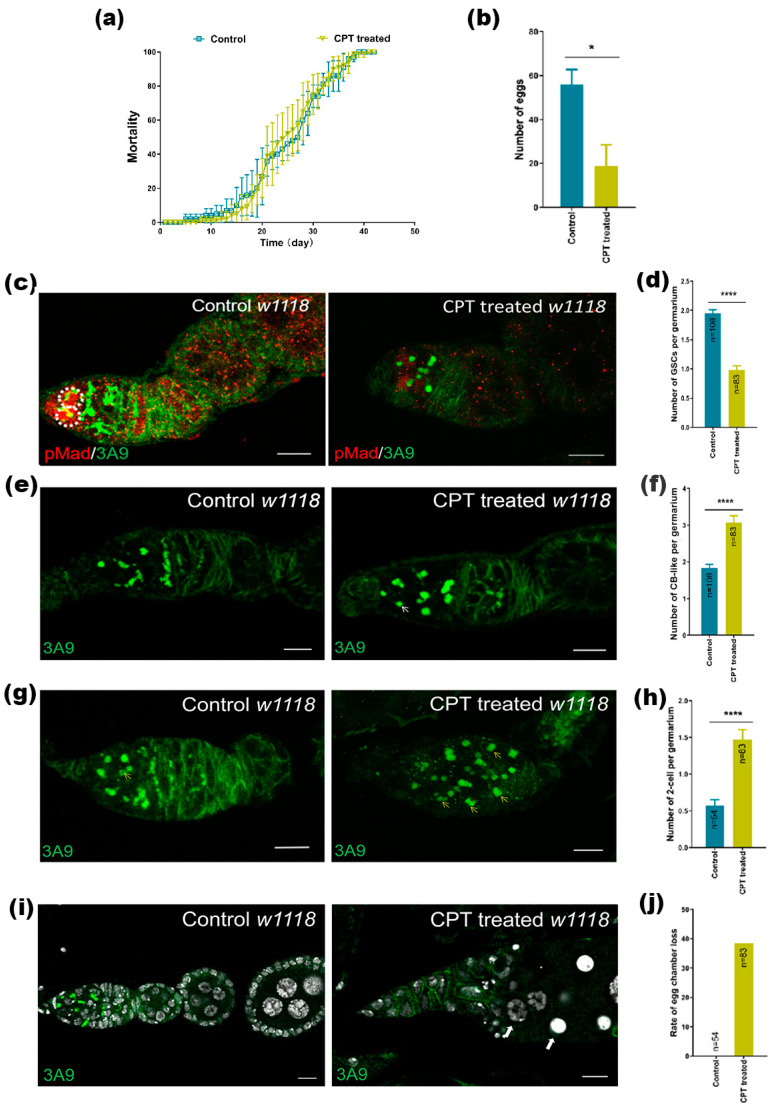
The effects of CPT on germ cells. (**a**) The mortality of flies after the flies were treated by CPT. (**b**) The number of eggs laid at the 9th d post CPT or DMSO solution treatment. (**c**) GSCs stained with anti-pMad and anti-3A9 antibodies in germarium at the 9th day post CPT or DMSO treatment. GSCs were indicated by white dashed circle. (**d**) Statistical data showing the GSC number from control and treated groups. (**e**) Germ cells stained with anti-3A9 antibody in germarium at the 9th day post CPT or DMSO treatment. CB-like cells were indicated by white arrow. (**f**) Statistical data showing the CB number from control and treated groups. (**g**) Representative DMSO- and CPT-treated images showing the effect of CPT on two-cell stained with anti-3A9. two-cell was indicated by yellow arrow. (**h**) Statistical data showing the two-cell number from control and treated groups. (**i**) Representative control and CPT-treated images showing CPT treatment blocked germ cell differentiation leading to accumulation of CB and two-cells and stained by Hoechst (white) and 3A9 (green). The white arrow indicated egg chamber loss. (**j**) The number of egg chambers showing abnormal phenotype, including apparent failure of egg chamber budding leading to empty egg chamber. Values are the means (±SEs) of replicates. Statistical comparisons were based on Students’ s *t*-tests. The level of significance for the results was set at * *p* < 0.05, **** *p* < 0.0001. Scale bar, 10 μm.

**Figure 2 ijms-24-01617-f002:**
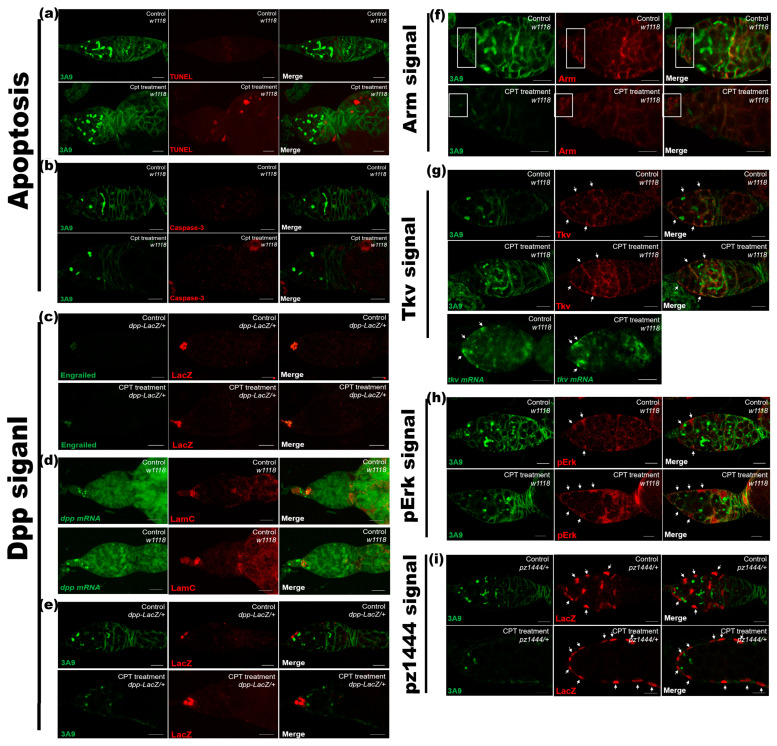
The roles of apoptosis and niche in CPT-induced GSCs loss. (**a**) Apoptotic signals from ovaries of control and CPT treated flies labeled with TUNEL (red) and stained with anti-3A9 (green). (**b**) Caspase-3 activity in CPT-treated and control germarium staining by Hoechst (white), Caspase-3 (red), and anti-3A9 (green). (**c**) Control and CPT-treated germaria with anti-En (green) and anti-LacZ (red) staining. (**d**) In situ hybridization examining *dpp* expression with anti-LamC (red) staining. (**e**) Control and CPT-treated germaria with anti-LacZ (red) and anti-3A9 (green) staining. (**f**) Arm immunoprecipitation with or without CPT exposure with Arm (red) and anti-3A9 (green) staining. Meanwhile, square flame indicated the expression of Arm. (**g**) Control and CPT-treated germaria with anti-Tkv (red) and anti-3A9 (green) staining, and in situ hybridization showing *tkv* mRNA expression in control and CPT-treated germaria. (**h**) Control and CPT-treated EC with anti-pErk (red) and anti-3A9 (green) staining. White arrow indicated the signal of pErk. (**i**) Control and CPT-treated EC with anti-LacZ (red) and anti-3A9 (green) staining. White dashed circle indicated ectopic location of ECs. Newly emerged flies (1 day post emergence) were treated with 100 mg/L CPT and control flies received equal amount of DMSO. The flies were dissected at the 9th day post treatment. Scale bar, 10 μm.

**Figure 3 ijms-24-01617-f003:**
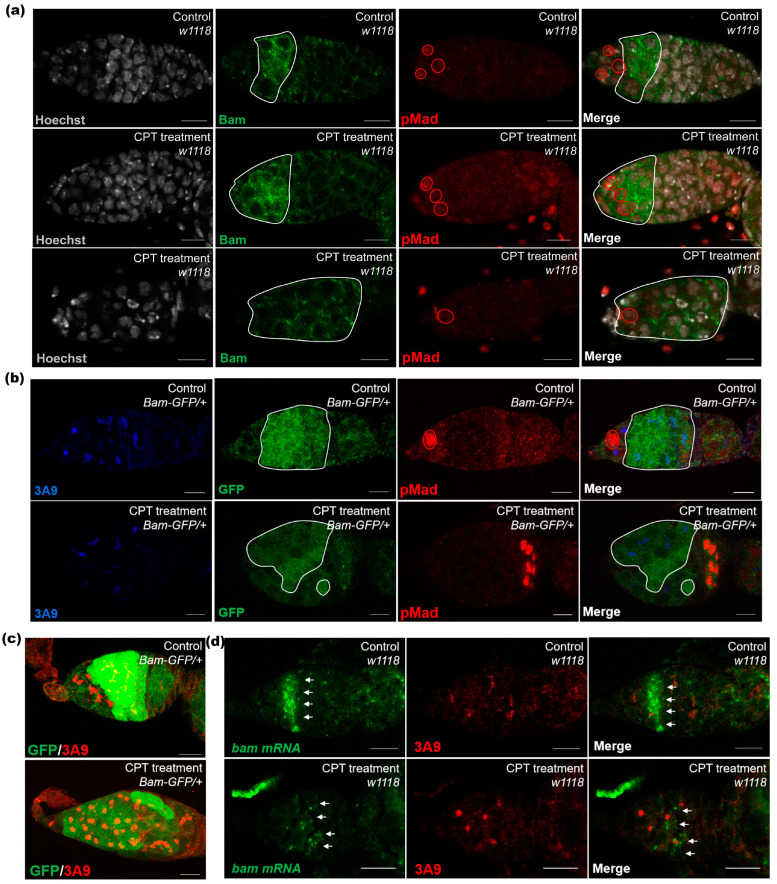
CPT could influence differentiation process by regulating Bam expression. (**a**) Control and CPT-treated germaria were labeled with Hoechst (white), anti-Bam (green) and anti-pMad (red) to examine Bam expression. White dashed circle indicated the location of Bam in control and CPT-treated germarium. (**b**) Germaria of female flies expressed GFP under the control of transcriptional of Bam stained with GFP (green), anti-3A9 (blue), and pMad (red). (**c**) Germaria of female flies expressed GFP under the control of transcriptional of Bam stained with GFP (green) and anti-3A9 (red). (**d**) In situ hybridization to examine *bam* expression at mRNA level with anti-3A9 (red) labeling with or without CPT treatment, white arrows indicated *bam mRNA* signal. Scale bar, 10 μm.

**Figure 4 ijms-24-01617-f004:**
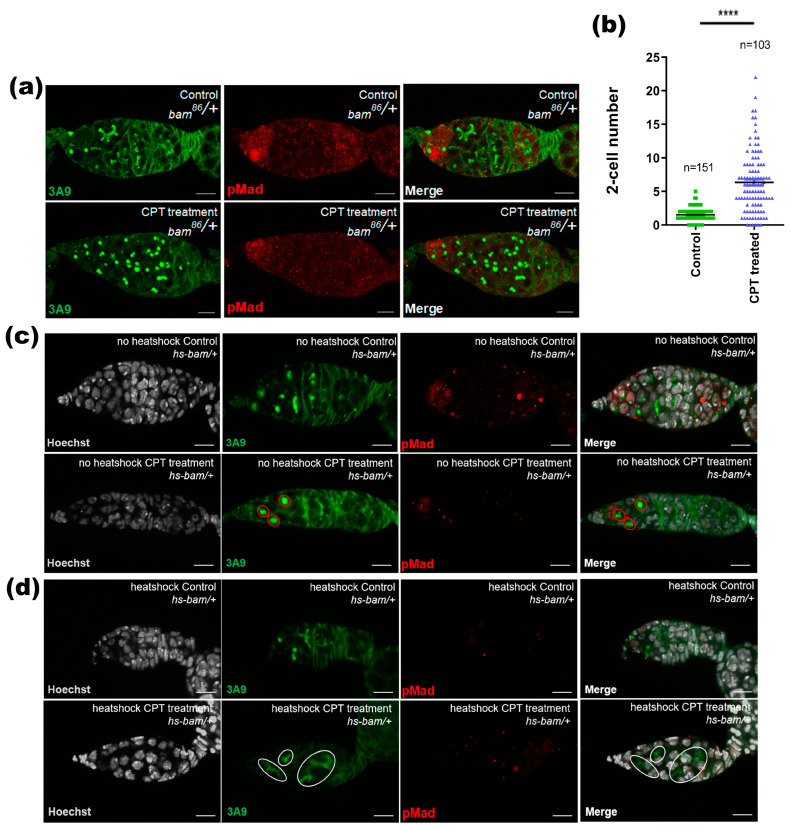
Bam can significantly rescue the two-cell accumulation caused by CPT. (**a**) *bam^86^* with anti-pMad (red) and anti-3A9 (green) staining with or without CPT treatment. (**b**) Data summarized the number of two-cell cysts in (**a**), green indicated control, blue indicated CPT treatment. (**c**,**d**) Control and CPT-treated female flies with or without exposure to heat shock to drive *bam* expression and staining with anti-pMad (red) and anti-3A9 (green). Statistical comparisons were based on Students’ s *t*-tests. The level of significance for the results was set at **** *p* < 0.0001. Scale bar, 10 μm.

**Figure 5 ijms-24-01617-f005:**
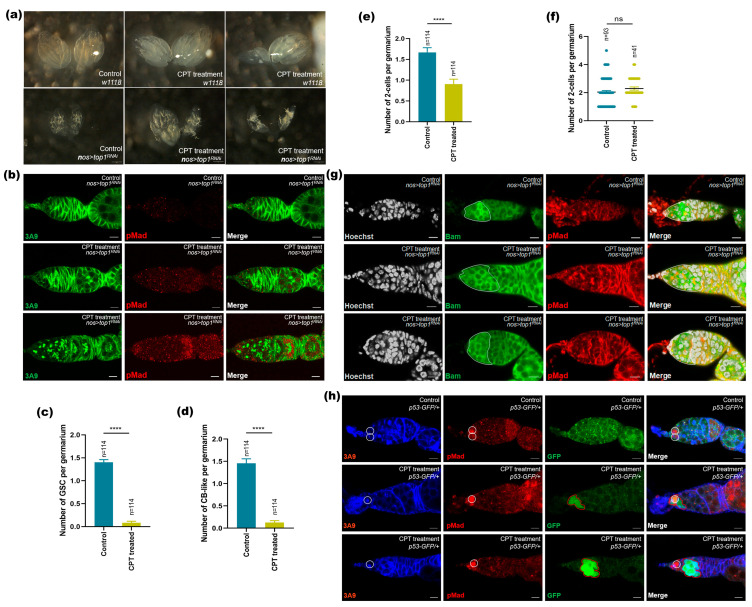
Top1 was involved in CPT-induced toxicity in germ cells. (**a**) Ovarian morphology from control and *top1*^RNAi^ germarium with or without CPT treatment. (**b**) Control and *top1*^RNAi^ germarium with or without CPT treatment and stained with anti-pMad (red) and anti-3A9 (green). (**c**) GSC quantification results are shown. (**d**) CB-like quantification results are shown. (**e**) Two-cell cysts quantification results are shown. (**f**) Two-cell cysts quantification results are shown by excluding the cases with germ cell loss. n is the number of the examined germaria; all the error bars represent SEMs; *p* values were calculated by comparing between *top1*^RNAi^ germarium with or without CPT treatment using Student’s *t* test. Blue indicated control, yellow indicated CPT treatment in (**c**–**f**). (**g**) Bam expression pattern in *top1*^RNAi^ germarium with or without CPT treatment staining with anti-3A9 (red) and anti-Bam (green). (**h**) Germaria of female flies expressed GFP under the control of transcriptional of p53 stained with anti-3A9 (blue), anti-GFP (green), and anti-pMad (red). White dash circle indicated GFP signals. The level of significance for the results was set at **** *p* < 0.0001. Scale bar, 10 μm.

**Figure 6 ijms-24-01617-f006:**
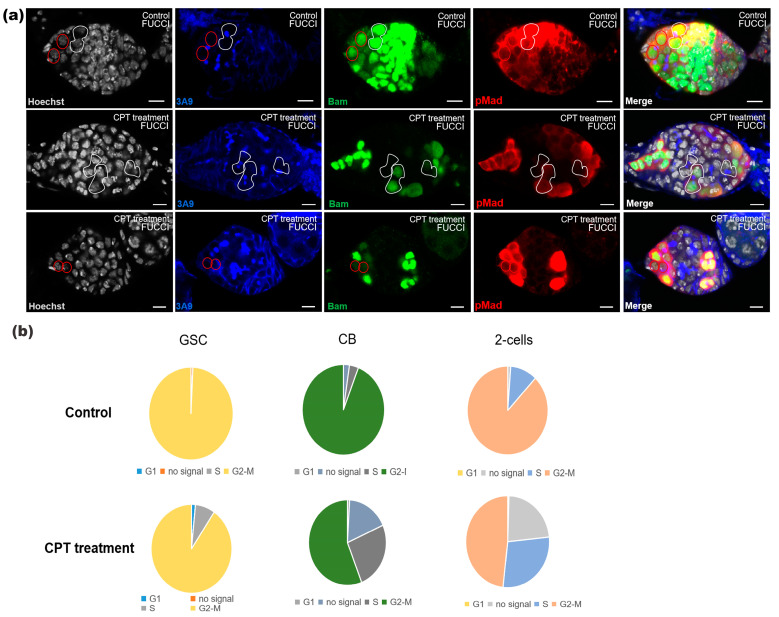
Cell cycle was determined by FUCCI system. (**a**) FUCCI were applied to identify cells in different phases of the cell cycle and stained with Hoechst (white), anti-3A9 (blue), anti-GFP (green), and anti-RFP (red). White dashed circle indicated two-cell without signals. (**b**) Quantification of cells in different phases of cell cycle.

**Figure 7 ijms-24-01617-f007:**
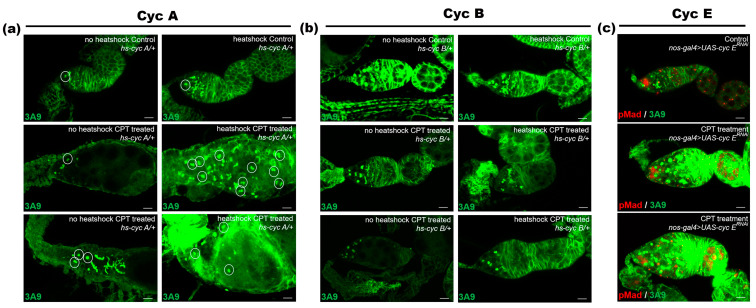
CycA was involved in CPT-induced GSC loss and two-cell accumulation. (**a**) Control and CPT-treated female flies with or without exposure to heat shock to drive *CycA* expression and staining with anti-3A9 (green). (**b**) Control and CPT-treated female flies with or without exposure to heat shock to drive *CycB* expression and staining with anti-3A9 (green). (**c**) Control and *CycE*^RNAi^ germarium and stained with anti-pMad (red) and anti-3A9 (green).

**Figure 8 ijms-24-01617-f008:**
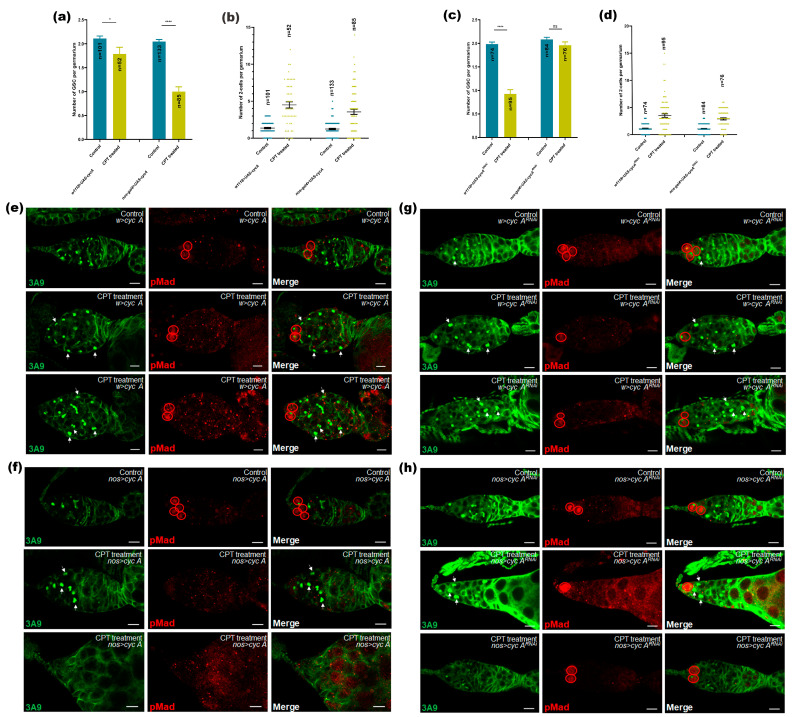
CycA is involved in GSC loss and two-cell accumulation induced by CPT treatment. (**a**) The GSC number of control and *CycA* overexpression germaria. (**b**) The number of two-cell cysts in control and *CycA* overexpression germaria. (**c**) The number of GSCs in control and *CycA*^RNAi^ germaria. (**d**) The number of two-cell cysts in control and *CycA*^RNAi^ germaria. (**e**,**f**) Control and *CycA* overexpression germaria with or without CPT treatment and stained with anti-pMad (red) and anti-3A9 (green). Blue indicated control, yellow indicated CPT treatment in (**c**–**f**). (**g**,**h**) Control and *CycA*^RNAi^ germaria with or without CPT treatment and stained with anti-pMad (red) and anti-3A9 (green). Statistical comparisons were based on Students’s *t*-tests. The level of significance for the results was set at * *p* < 0.05, **** *p* < 0.0001. Scale bar, 10 μm.

## Data Availability

All data generated or analyzed during this study are included in this published article and its Appendix A. Reagents used in this publication will be provided upon request.

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
