# Peer review of "Camptothecin Effectively Regulates Germline Differentiation through Bam–Cyclin A Axis in Drosophila melanogaster"

_ijms, 2023, doi:10.3390/ijms24021617_

Round 1

Reviewer 1 Report

This paper reveals the mechanism of camptothecin (CPT) to exert its anticancer activity, provides important data on its toxicological role, and the results are convincing, suggesting the therapeutic potential of CPT as an anticancer agent for germ cells. I agree to the publication of the manuscript.

Author Response

This paper reveals the mechanism of camptothecin (CPT) to exert its anticancer activity, provides important data on its toxicological role, and the results are convincing, suggesting the therapeutic potential of CPT as an anticancer agent for germ cells. I agree to the publication of the manuscript.

Response: We thank for this reviewer’s good comments.

Reviewer 2 Report

In the current study the Authors present the investigations concerning the influence of CPT in GCS differentiation.

I think the study is well planned and the results are clearly presented. 

The authors may consider shortening chapter 2. Results, while expanding chapter 3.

Is it possible to provide micrographs in better resolution?

Author Response

 In the current study the Authors present the investigations concerning the influence of CPT in GCS differentiation. I think the study is well planned and the results are clearly presented

Major comments:

  1. The authors may consider shortening chapter 2. Results, while expanding chapter 3.

Response: We thank this reviewer for his/her suggestion. We added detailed descriptions underlying the role of top1 in CPT treatment in chapter 3. And for chapter 2, the results part, we tried to present detailed descriptions for our results, so that the readers could get clear picture for our study. We thank this reviewer for his/her kind suggestion.

  1. Is it possible to provide micrographs in better resolution?

Response: We thank this reviewer for his/her suggestion and we are sorry for our oversight. We have updated the images with high resolution.

Reviewer 3 Report

Jing Zhang et al have identified that Bam- cyclin A axis is involved in the Camptothecin mediated germline stem cell loss and 2-cell cysts differentiation defects by arresting cell cycle in Drosophila. Also, the authors show that Top1 inhibition facilitates the camptothecin mediated GSC loss. The strength of the report is the use of several approaches to identify the underlying toxicological effects of anti-cancer drug Camptothecin in reproductive system and urges to validate its toxicity in advanced models.However, the following issues needs to be addressed. 

-       Authors claim that inhibition of top1 (top1RNAi) shrink the germarium and led to rapid germ cell loss, and CPT treatment worsens GSC loss phenotype and CB loss (Fig. 5a&b). But no direct evidence has been shown. Author should show whether CPT treatment inhibits top1 or overexpression of top1 leads to rescuing CPT mediated GSC loss phenotype and CB loss.

-       Authors shows CPT treatment induces upregulation of p53 activities and DNA damage in GSC as well as in 2-cells in CPT-treated germarium (Fig. 5h) and concludes CPT can cause DNA damage in early germline cell by regulating Bam expression at both transcript and protein level, thus leading to cell arrest at G1/S and 2-cell accumulation. It would be appropriate if author show the yH2Av foci (DNA damage marker) in those conditions.

Minor comments-

-       Poor figure quality and increase fond size in Figures including graphs.

-       Missing error bars and significance in figure 1J

Author Response

Jing Zhang et al have identified that Bam- cyclin A axis is involved in the Camptothecin mediated germline stem cell loss and 2-cell cysts differentiation defects by arresting cell cycle in Drosophila. Also, the authors show that Top1 inhibition facilitates the camptothecin mediated GSC loss. The strength of the report is the use of several approaches to identify the underlying toxicological effects of anti-cancer drug Camptothecin in reproductive system and urges to validate its toxicity in advanced models.However, the following issues needs to be addressed.

Major comments:

  1. Authors claim that inhibition of top1 (top1RNAi) shrink the germarium and led to rapid germ cell loss, and CPT treatment worsens GSC loss phenotype and CB loss (Fig. 5a&b). But no direct evidence has been shown. Author should show whether CPT treatment inhibits top1 or overexpression of top1 leads to rescuing CPT mediated GSC loss phenotype and CB loss.

Response: We thank this reviewer for his/her suggestion. As this reviewer kindly suggested, our results indicated that top1RNAi flies with CPT treatment could exacerbate GSC loss phenotype induced by CPT treatment alone, exhibiting more severe abnormal germarium and triggering almost entire GSC and CB loss. After we performed RNAi against top1 with CPT treatment, we also tried to overexpress top1 to finally confirm the role of top1 in CPT-induced GSC loss as suggested this reviewer. However, it seems hard to overexpress top1 upon CPT treatment to rescue phenotypes as top1-cleavable complexes results in transcription arrest and ‘irreversible’ strand breaks, which seems to be difficult to be rescued by overexpression. And the inhibitory effect of CPT on the expression of top1 has been well-documented in a large number of literatures, as Eng and Nitiss have proved that CPT can effectively inhibit the expression of top1 [1, 2]. We thank again for such good suggestion, and we added detailed discussion regarding this in the revised version.

  1. Eng WK, Faucette L, Johnson RK and Sternglanz R. Evidence that DNA topoisomerase I is necessary for the cytotoxic effects of camptothecin. Mol Pharmacol 1988; 34: 755-760.
  2. Nitiss J and Wang JC. DNA topoisomerase-targeting antitumor drugs can be studied in yeast. Proc Natl Acad Sci U S A 1988; 85: 7501-7505

  1. Authors shows CPT treatment induces upregulation of p53 activities and DNA damage in GSC as well as in 2-cells in CPT-treated germarium (Fig. 5h) and concludes CPT can cause DNA damage in early germline cell by regulating Bam expression at both transcript and protein level, thus leading to cell arrest at G1/S and 2-cell accumulation. It would be appropriate if author show the γH2Av foci (DNA damage marker) in those conditions.

Response: We thank this reviewer for his/her suggestion. As this reviewer kindly suggested, we performed additional experiment and provided the immunostaining results of γH2Av (DNA damage marker) under the treatment of CPT. We used rabbit-anti-γ H2Av (1:2000) further determined whether CPT could cause DNA damage to GSC and 2-cells. The results showed that GSC and CB exhibited no significant signals, indicating that DNA double strand breaks did not occur under CPT treatment (Fig.1).

We raised reasons for the lack of signals: since H2Av is a marker of DNA double strand break (DSB), and mutants defective in DNA damage tolerance are susceptible to prolonged replication fork stalls and fork collapse following exposure to DNA-damaging agents, resulting in DSB and genome instability. While CPT could prevent the re-ligation of the nicked DNA and dissociation of top1 from the DNA by binding to both of the top1 enzyme and the intact DNA strand through hydrogen bonding. During replication, this CPT-involved ternary complex could act as a roadblock for the replication fork to result in shear stress upon the intact DNA strand, and eventually leading to breakage, DNA damage, and cell death. CPT did not belong to DNA damage reagent, and the strain we used was wild type, thus, H2av staining did not produce significant signals in early germ cells. In addition, the up-regulation of p53 is a sign of inducing DNA damage, which we presented in our results. In our study, CPT treatment on the 9th day may be the first stage of inducing DNA damage.

We thank this reviewer for his/her good suggestions, however, this study mainly focused on the regulation of cell cycle induced by CPT on early germ cells through Bam-Cyc A, resulting in defects in GSC maintenance and 2-cell accumulation. We are still studying how top1 can induce DNA damage, leading to the mechanism of periodic changes, which we wish to present in our further study.

Minor comments:

  1. Poor figure quality and increase fond size in Figures including graphs.

Response: We thank this reviewer for his/her suggestion and we are sorry for our oversight. We have updated the images with high resolution.

  1. Missing error bars and significance in figure 1J.

Response: We thank this reviewer for his/her suggestion. Picture 1J shows the frequency of egg chamber loss after CPT treatment, so there is no error bar and difference analysis.

Round 2

Reviewer 3 Report

No further comments

Author Response

Thank you.